# Barriers and Opportunities Regarding Implementation of a Machine Learning-Based Acute Heart Failure Risk Stratification Tool in the Emergency Department

**DOI:** 10.3390/diagnostics12102463

**Published:** 2022-10-11

**Authors:** Dana R. Sax, Lillian R. Sturmer, Dustin G. Mark, Jamal S. Rana, Mary E. Reed

**Affiliations:** 1Kaiser Permanente Northern California Division of Research, Oakland, CA 94612, USA; 2Department of Emergency Medicine, The Permanente Medical Group, Oakland, CA 94612, USA; 3College of Osteopathic Medicine, Touro University, Vallejo, CA 94592, USA; 4Department of Cardiology, The Permanente Medical Group, Oakland, CA 94612, USA

**Keywords:** emergency medicine, acute heart failure, risk stratification, clinical decision support, machine learning, predictive models, qualitative research, implementation, workflow integration

## Abstract

Hospital admissions for patients with acute heart failure (AHF) remain high. There is an opportunity to improve alignment between patient risk and admission decision. We recently developed a machine learning (ML)-based model that stratifies emergency department (ED) patients with AHF based on predicted risk of a 30-day severe adverse event. Prior to deploying the algorithm and paired clinical decision support, we sought to understand barriers and opportunities regarding successful implementation. We conducted semi-structured interviews with eight front-line ED providers and surveyed 67 ED providers. Audio-recorded interviews were transcribed and analyzed using thematic analysis, and we had a 65% response rate to the survey. Providers wanted decision support to be streamlined into workflows with minimal disruptions. Most providers wanted assistance primarily with ED disposition decisions, and secondarily with medical management and post-discharge follow-up care. Receiving feedback on patient outcomes after risk tool use was seen as an opportunity to increase acceptance, and few providers (<10%) had significant hesitations with using an ML-based tool after education on its use. Engagement with key front-line users on optimal design of the algorithm and decision support may contribute to broader uptake, acceptance, and adoption of recommendations for clinical decisions.

## 1. Introduction

Emergency department (ED) risk stratification and identification of lower-risk patients with acute heart failure (AHF) who may be amenable to safe outpatient care continues to present a significant challenge. There are over one million ED visits across the United States each year for AHF, and 80% of patients are admitted to hospital [1]. Hospital admissions and readmissions continue to increase [2], and total annual costs for heart failure are estimated to reach $70 billion by 2030, with 80% being due to hospital admission [3,4]. 

ED providers play a crucial role in the initial stabilization, management, and disposition decision-making for patients with AHF. Accurate risk stratification of ED patients with AHF is challenging due to the medical and social complexity of these patients. Studies demonstrate the challenge of optimal alignment of resource intensity with patient risk, with many lower-risk patients being admitted to hospital, higher-risk patients being discharged [5,6], and high rates of adverse events among discharged patients [5,6,7,8,9]. These significant discrepancies between patient risk and hospital admission decisions highlight the impetus for development and testing of risk prediction tools. Recently, several ED risk stratification tools for AHF have been developed and are in various stages of validation and evaluation [6,10,11,12,13]. 

Our study team recently developed a machine learning (ML)-based AHF risk prediction tool to predict the risk of a 30-day severe adverse event [6]. As we prepare to implement this tool into real-time clinical workflows, we plan to assess feasibility (including technical aspects of electronic health record (EHR) build, score calculation timing, handling of missing data, and clinician-facing display), acceptability (provider use of tool and adoption of recommendations), and utility (impact of tool on key clinical outcomes). 

Implementation of a new risk tool and clinical decision support (CDS) into a busy ED can be challenging [14,15,16]. EHR tools for patients with AHF have been successfully deployed, with limited impacts observed when decision support was not added to prognostic scores [17,18]. Prior studies highlight an underappreciation of the steps needed to translate model predictions into real-time use and improved health [19,20]. Successful implementation requires careful consideration of various factors, including workflow adaptability [14], technical infrastructure [15,21], and clinician acceptance [14,15,22,23,24]. Some studies suggest lack of trust or familiarity with ML models may make their implementation more challenging [22,25]. 

While several studies describe the importance of establishing a framework for risk tool implementation [19,20,21,22,23,26,27], to our knowledge, none have focused on ED physician-reported feedback to guide the planned implementation process of a ML-based tool paired with CDS in the ED setting. Moreover, without careful integration into the existing workflow, the tool and CDS may not reach their full potential and may even negatively impact patient care [28,29]. The goal of this study was to explore factors influencing the integration and adoption of a ML-based risk prediction tool and paired CDS for ED patients with AHF into clinical workflows. To achieve this goal, we conducted semi-structured interviews and surveys with front-line ED physicians and used a mixed-methods analysis approach to better understand barriers and opportunities regarding optimal implementation of the tool and paired CDS.

## 2. Material and Methods

### 2.1. Design and Study Setting

Using a mixed-methods approach and conceptual model based on the Consolidated Framework for Implementation Research [19], we conducted semi-structured, in-depth video interviews with 8 ED providers and surveyed 103 ED providers. Participants were from The Permanente Medical Group, a physician group with more than 9000 physicians. Kaiser Permanente Northern California (KPNC) is an integrated healthcare delivery system serving more than 4 million members in 21 hospital-based medical centers. We studied ED providers who work at 3 of the KPNC community EDs that are part of this large integrated delivery system. This study was approved by the KPNC Institutional Review Board. 

### 2.2. Participants

We recruited emergency physician participants for the interviews using snowball sampling, aiming to include variation in years of practice, gender, and clinical versus operational roles within the department [30]. All full-time practicing physicians at these sites were sent the survey electronically. Emergency physicians were notified about the study during a department meeting and by electronic communication and were subsequently invited to partake in the interviews and enroll in the survey via email and an opt-in process.

Participants provided verbal consent before taking part in the interview and online consent for the survey. Interview participants received a small compensation for their participation. Interviews were between 30–60 min in length. This study follows the Consolidated Criteria for Reporting Qualitative Research (COREQ) [31] and Standards for Reporting Qualitative Research reporting guidelines [32].

### 2.3. Interview and Survey Instruments 

The study team, including 2 practicing emergency physicians and 1 health services researcher, developed the interview guide (available in Appendix A). Interview questions were open ended and designed to elicit descriptions from participants related to optimizing implementation of an electronic ML-based AHF risk prediction tool and paired CDS. We piloted the interview guide with 2 ED providers to identify areas for refinement prior to initiating data collection. The provider interview guide included domains on general adoption of decision support in the ED, clinical areas where CDS would be most useful, hesitations about use of ML-based risk models, and opportunities to improve acceptance and adoption. The semi-structured format allowed new topics to emerge during interviews.

Based on responses from the interviews, we developed a survey instrument (see Appendix A). The survey allowed us to collect data on a larger sample of eventual end-users and probe similar questions as those in the physician interviews using quantitative (categorical and ordinal, via Likert scale) responses. The survey instrument was piloted by 2 ED physicians, and iterative changes were made prior to sending to all 103 ED physicians at the study sites. 

### 2.4. Data Collection

The PI for this project, a practicing emergency physician and clinical researcher, conducted all interviews between June and July of 2021 via Microsoft TEAMS (Microsoft Corporation, Redmond, WA, USA). Interviews were 30–60 min in length, and audio was recorded and then transcribed into a written transcript. The PI wrote field notes after interviews on findings and emerging themes, and the study team met biweekly during study collection to discuss themes and determine when thematic saturation was reached. The survey was sent electronically to ED providers in August of 2021, and participants were given 3 reminders to complete the survey over a 1-month period. 

### 2.5. Qualitative Analysis

We used a thematic analysis approach to coding and analysis of interview data, including both inductive and deductive coding [31,33,34]. We summarized individual transcripts and developed a summary table with broad themes (use of risk stratification tools in practice, areas where CDS would be most helpful for ED patients with AHF, and barriers and opportunities regarding increased adoption), sub-themes (risk tolerance, alerts for tool use and CDS display, and medico-legal concerns), and illustrative quotes. 

We used a rapid analysis process for analyzing the data collected through the physician interviews [35]. This allowed for timely results and maximized effective implementation. Prior studies have shown that a rapid analysis process efficiently provides pragmatic and actionable findings [35,36]. For survey data, we used descriptive statistics to summarize survey data in numerical and visual form. 

## 3. Results

Interview participants varied in their demographic characteristics (25% were female, 38% were non-white); years since completing training (5–30 years); and involvement in clinical, administrative, and research activities. A total of 67 participants responded to the survey, representing 65% of the emergency physicians who received the electronic survey. Among surveyed physicians, 39% were female; 40% were non-white; and years since completing residency training varied from 2 to 30, with a median of 12 years. 

Interviewed providers raised several over-arching themes that may optimize or detract from successful implementation of ML-based AHF risk models and CDS into real-time practice. We have broadly separated these into three domains, ordered by increasing specificity to our HF risk tool: (1) general considerations for implementation of risk tools and CDS in the ED (Table 1), (2) AHF-specific risk tool and CDS implementation considerations (Table 2 and Figure 1), and (3) barriers and opportunities regarding successful implementation of ML-based AHF prediction models (Table 3). For each domain, we delineate corresponding subthemes and highlight illustrative quotes.

### 3.1. General Implementation and Use of CDS Tools

Nearly all surveyed and interviewed ED providers routinely use risk tools and CDS in their practice, with 90% reporting that they use CDS at least 1–2 times a week, and 56% using CDS at least 1–2 times a shift. Providers note they use CDS specifically when it helps with a management issue, particularly for assistance with admission-related decision- making. Providers stressed the importance of agreement on CDS recommendations and care protocols with consulting services. Table 1 presents themes and key quotes about ED providers’ views on usefulness of CDS, hesitations about CDS use, and opinions on optimal prompts to access CDS.
diagnostics-12-02463-t001_Table 1Table 1Themes and representative quotes from qualitative interviews related to general use of risk prediction tools and CDS in clinical practice in the ED.Theme Key PointsQuotes**Use of risk tools and decision support**• Most physicians use risk tools in their practice, often to address a specific management issue. • “I usually use decision support tools to check myself, to see if I’m being too blasé; it can be like a safety check.”**Hesitations with using risk tools**• There were concerns about applying risk tools for the right patient and setting• Admission CDS less likely to be used if admitting team did not follow the same risk-based pathways.• “Sometimes two services use two different tools and then you get two different outcomes, which feels uncomfortable. It’s important to have buy in from the other departments that sometimes don’t have the same perspective… Even within a department, there’s a spectrum of risk tolerance.”• “Unless it’s helping with a disposition or management step, I don’t use it.”**Where do ED physicians most frequently access decision support?**• Most physicians used external platforms, especially MDCalc• Many also used internal platforms, including a home-grown clinical decision support platform (“RISTRA”), internal department websites, and cues within the electronic health record (such as smart phrases and ordersets)• “RISTRA (a KPNC-designed platform for decision support) is kind of a win-win in terms of having the documentation component that I can bring into the chart.”


### 3.2. AHF-Specific Risk Stratification and CDS Considerations

Table 2 and Figure 1 present AHF-specific considerations regarding using risk tools and CDS, with representative quotes from interviews. Respondents wanted predictive models to assist with clinically challenging areas, including ED and outpatient medication adjustments and disposition decisions. Many also stressed that use of risk models and standardized risk-based pathways may provide medico-legal protection. Providers emphasized they wanted to be presented with relevant risk information specifically when it would be used for clinical decision making.
diagnostics-12-02463-t002_Table 2Table 2Themes and representative quotes from qualitative interviews related to use of AHF risk prediction tools and CDS.ThemeKey PointsQuotes**Risk prediction**• CDS should primarily assist with risk prediction and disposition decisions. • “I think that it would be useful to have a scoring system that I could refer to that would help me with the discussion with my admitting hospitalist.”• “A summary risk statement that we can cut and paste into our note would be helpful… for example, ‘Having reviewed your chart and multiple variables in your care, your risk is X, and we feel comfortable discharging you.’” **Which patients with AHF are the most difficult to risk stratify?**• Assistance would be most useful for stratifying the low- and moderate-risk patients.• “When the oxygen saturation is peri−90 is when I would love a little extra decision support… will the person do OK if I send them home with oxygen or should I admit them? Or observe them?” • “The middle-of-the-range patients… I think most of us that have been practicing for a little bit, we kind of know who belongs in the ICU. And we are also pretty sure about which ones can go home.”**Risk tolerance for discharging patients**• Risk tolerance and acceptable ‘adverse event rates’ for patients being discharged with AHF is variable.• Having standardized care pathways with accepted risk thresholds would be useful to improve care and provide medico-legal protection.• “It would be helpful to refer to our system’s ‘standard of care.’ Then I would feel more comfortable discharging a patient.”• “I think it would be key to normalize the risk estimates—display what the average risk is for ED patients with AHF in KPNC, or nationally, so the provider and patient have some context.”• “I think a risk estimate would help both for medico-legal protection for patients I want to discharge and for help convincing the hospitalist for patients I want to admit.”• “If there’s a big school of fish, as long as you are contiguous with the school, I think you’re fine. You don’t want to be an outlier to the school and be the obvious one to be picked up by the predator.”**Optimal display of risk estimates**• Risk displays should include both the patient’s risk class and precise 30-day risk estimate.• “I think it would be helpful to have both [specific number and risk class]… I think when explaining to patients, it generally would be helpful to have a category, and if they ask follow-up questions you can actually give them a number if it seems like something that they can comprehend.” • “It would be nice if the risk categories were color coded.”**Additional uses for AHF CDS**• Personalized medical management, sign-out plans for oncoming providers, and discharge plans among patients going home.• “For patients that I’m discharging, if there was some additional support to say ‘this is how you should titrate your diuretics, and this is how you should adjust your blood pressure medication’, that would be helpful.”• “It would really help with sign out. For example, if I’m signing out in one hour, I can say to the oncoming MD, ‘according to the tool, we are giving this specific dose of Lasix and blood pressure control’; if in three hours the targets haven’t been met, then the patient should be admitted. That would be so useful because then I can sign out comfortably.”• “I think having a grid with conversion doses between IV and PO Torsemide and Bumex… and titration of diuretics and timing of outpatient labs if plan is to discharge would be helpful.”• “It would be helpful if it linked to specific discharge instructions, like ‘check your weight, follow up with your doctor, monitor your diet, adjust your diuretics, etc.… it would just feel like a coherent and clear plan with standardization across physicians.”• “I would like some education on which patients to include, about the differences in use for patients with preserved versus reduced ejection fraction, and how to incorporate data from an echo report (what values to look for).”**Optimal prompts to use an AHF CDS**• Providers only want alerts/reminders if they are for the right patient at the right time.• “I think most docs will want a nudge rather than having to just remember to click on that risk tool.”• “I would not want an alert until I’m ready to use it. So, if the score relies on labs [troponin and BNP], I don’t want an alert until the labs are back.”• “I don’t want an alert on someone I won’t use the risk score for… someone who is hypoxic and clearly sick and being admitted—an alert to use a risk score will just be an annoyance.” 


### 3.3. ML-Based AHF CDS

Table 3 and Figure 2 present ED provider perspectives on using an ML-based risk model for ED patients with AHF. Among all surveyed physicians, only 8% disagreed or strongly disagreed that they would feel comfortable using an ML-based model to assist with decision-making after they were first educated on how the model worked. Many physicians recognized the benefits of ML-based models but expressed some fear using something they did not fully understand (concern over what variables were included and how the algorithm works). Clinicians would be more comfortable if the models were biased conservatively to decrease the risk of serious adverse events. Receiving education on how the risk tool and CDS works and feedback on patient outcomes after use were seen as opportunities for increasing acceptance.
diagnostics-12-02463-t003_Table 3Table 3Themes and representative quotes from qualitative interviews with physicians related to opportunities to increase comfort using an ML-based AHF risk model.ThemeKey Points Quotes**Comfort with using an ML-based model**• Providers noted mild-to-moderate hesitation with using ML-based models to assist with decision making. This hesitation could be mitigated by education and by data showing that the model’s predictions and recommendations were safe. • “If there were some way for me to click to find out more information about what’s going on behind the scenes, that would be helpful.”• “I think I would be slightly more hesitant [to use an ML model compared to a simpler model]… The discomfort would be mitigated if it were to bias towards a more conservative pathway of care.”• “I wouldn’t care if it’s a black box. I would have moderate hesitation to start, but it would go to mild after some education and if it were to lean towards more conservative practice.”• “I think machine learning can be quite useful. I think that the main thing is to know if it’s changing the recommendations over time as it gets more data.”**Concerns with using an ML-based model**• A few providers raised concerns about how an ML-based model might impact disparities in care. • “I’m curious how using ML models in clinical practice would or would not affect equity in various patient populations.”• “Using ML, it is very difficult to ensure that the outputs don’t reflect input biases during the creation or coding of the dataset. Additionally, since by definition, ML-models allow for non-linear functions and relationships, they are essentially impossible to interrogate from the outside and are opaque to the end user.”• “I would want to make sure that KPNC members vs non-members analysis is included in learnings and outcomes research.”**What would increase your likelihood of using an ML-based AHF risk model?**• Providers’ enthusiasm to use an ML-based risk estimates and CDS would increase if data showed CDS added to physician gestalt. Providers wanted education on risk models and CDS use and feedback on outcomes. • “It would be helpful to have an education session first… to understand how the model works and how to use the CDS.” • “If it were shown that this tool is both safer for patients and better for our organization, that would cause me to use it.”• “It would be helpful to get data, feedback, and reinforcement constantly from both sides [ED and admitting team] so we know it’s a functioning method for us to agree on how we’re dealing these patients.”• “Regular feedback, at least initially and whenever changes are made to the model, would be helpful and build trust in the model.”


## 4. Discussion

We present data from a large and diverse sample of ED physicians related to use of risk-based decision support for patients with AHF. Physicians from 3 community EDs were studied, with 8 participating in in-depth interviews and 64 participating in a survey. Our overarching goal was to collect front-line physician feedback to guide the implementation of an ML-based risk tool with paired CDS. Through the interviews and survey, we explored factors that might influence the adoption of the risk tool and CDS into real-time clinical workflows.

We probed physicians on their general acceptance of risk models and use of CDS, and specifically on when and why they might choose to adopt risk-based CDS for patients presenting with AHF. Providers mentioned several clinical areas where assistance would be most useful and described factors that would increase their acceptance of these tools. We asked specifically about hesitations regarding the use of ML-based models in clinical practice and strategies to mitigate these hesitations.

Two recent studies of patients with AHF prove that an EHR -based risk prediction tool can be deployed in real-time and highlight the potential benefits and limitations of risk predication and decision support. A novel, targeted EHR-based alerting system for outpatients with heart failure with reduced ejection fraction led to significantly higher rates of guideline-directed medical therapy at 30 days compared with usual care [17]. On the other hand, there were no significant differences in key patient outcomes (30-day hospitalization, 1-year mortality, or adherence to guideline-directed medical therapy) among patients who had a 1-year mortality risk prediction rate displayed to their inpatient providers [18]. This finding suggests that prognostic information alone (without paired decision support) does not change practice or outcomes. These studies highlight the value of key stakeholder engagement with CDS implementation and display of information to try to optimize its impact. 

ED providers in our study felt that personalized CDS would be most useful for venue of care decisions and, among patients suitable for safe outpatient care, guidance on a safe discharge plan. Patients with AHF are complex and heterogeneous, and often have multiple co-morbid illnesses and unique social needs. These factors complicate accurate risk stratification and admission-related decision-making. Interviewed providers noted that specific guidance for medium-risk patients would be most helpful, and that standardized risk-based protocols could help streamline conversations with admitting consultants and develop a local standard of care. ED providers are the main gatekeepers of hospitals and there is growing interest in considering alternate venues of care for appropriate patients [37,38]. Using risk models to accurately predict short-term prognosis may help support providers with these challenging decisions. 

Prior studies have demonstrated that there is often a mismatch between patient risk and admission decision, with frequent admissions of low-risk patients, and, simultaneously, frequent discharges of higher-risk patients [6,39]. A recent study using a large, multi-center, diverse population of ED patients with AHF suggested that use of risk-based protocols to assist with admission decisions can likely improve outcomes among discharged patients without leading to increased admissions [40]. It also identified a threshold that may be used to safely define low-risk patients (predicted 30-day mortality rate of 1% or less) for which hospitalization offered no perceptible survival benefit. 

ED providers also noted that personalized CDS to assist with the transition of care from ED to the outpatient setting would be useful. This may include medication reconciliation support at discharge, personalized self-care strategies, and follow-up planning. Recent U.S. and international guidelines give a Level 1A recommendation for four classes of medications for patients with heart failure with reduced ejection fraction, and Level 2a and 2b for two classes of medication for patients with heart failure with preserved ejection fraction. Tailored decision support for complex medication regimens and streamlined treatment protocols may help ensure patients are optimally medically managed. As close outpatient follow-up after an ED visit has been shown to reduce risk of short-term adverse events [5], personalized decision support on when and with whom discharged patients should follow up with may help promote safe discharge.

We asked providers about opportunities and barriers regarding acceptance and adoption of ML-based risk scores. As other studies have similarly found [27,41,42,43], providers wanted the right information displayed specifically at the right time. They considered a ‘positive user experience’ as the most important factor that would increase their use of AHF CDS. Specifically, they noted that alerts should be targeted and minimize unnecessary distractions, and that displayed clinical information must be streamlined into standard ED workflows. Providers noted various strategies to help nudge or alert them to use CDS in appropriate situations as well as visual cues and color coding to passively display risk. 

Some providers expressed concerns about using ML-based models because of their ‘black box’ nature and because of concerns surrounding propagating health inequities. Over 70% of surveyed providers agreed or strongly agreed on the need for education on how the model worked, and interviewed providers also asked that additional information on risk score development be available on demand. Providers felt that feedback on patient outcomes after CDS implementation showing safety of recommendations would increase their willingness to use CDS. There is growing concern that use of ML-based prediction models in healthcare may exacerbate disparities, and prior studies have shown disparate outcomes by patient race from algorithms for allocating healthcare resources [44,45,46,47,48]. A few providers in our study echoed these concerns, which might increase their hesitation to use the models. 

### Limitations

While our results highlight several key barriers and opportunities regarding successful implementation of an AHF risk tool, several limitations should be considered. KPNC is a unique healthcare system and its features (such as reliable outpatient follow up and comprehensive EHR) may not generalize to other practice settings planning to implement a risk tool. We only included providers from three EDs in one healthcare system, and their responses may not be generalizable to providers in different practice settings. We only conducted eight in-depth interviews, and this smaller sample size again limits generalizability of findings.

## 5. Conclusions

Our study findings from 8 interviewed and 67 surveyed ED physicians identify several important opportunities and barriers regarding implementation and real-time use of a ML-based risk model and CDS for ED patients with AHF. Providers noted that personalized decision support and use of risk-based pathways could help standardize hospital admission decisions and discharge planning. They highlighted the need for CDS to be integrated into workflows to minimize disruptions. Provider education prior to CDS implementation and regular feedback on patient outcomes were seen as opportunities for increasing acceptance and adoption. Lessons learned from this study will be used to optimize development and implementation of AHF CDS in our health-care system and can be applied to other health-care systems considering integrating ML-based models into their EHR. 

## Figures and Tables

**Figure 1 diagnostics-12-02463-f001:**
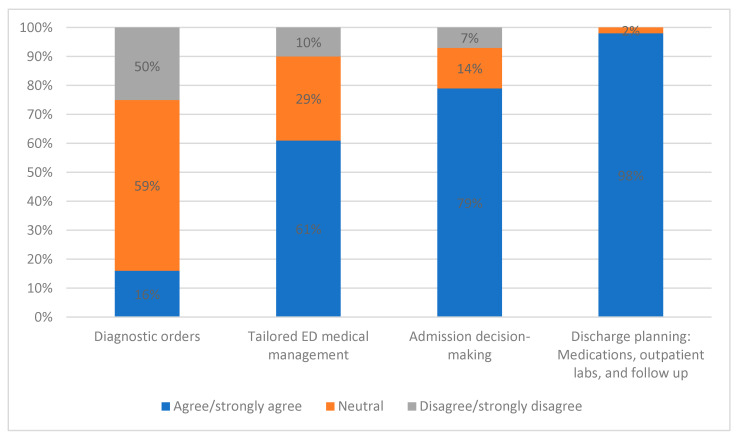
Survey Data: ED providers’ impressions of the clinical areas where personalized, risk-based CDS would be most useful. Bars represent percentage of providers responding “agree/strongly agree”, “neutral”, or “disagree/strongly agree” to each potential motivator of use.

**Figure 2 diagnostics-12-02463-f002:**
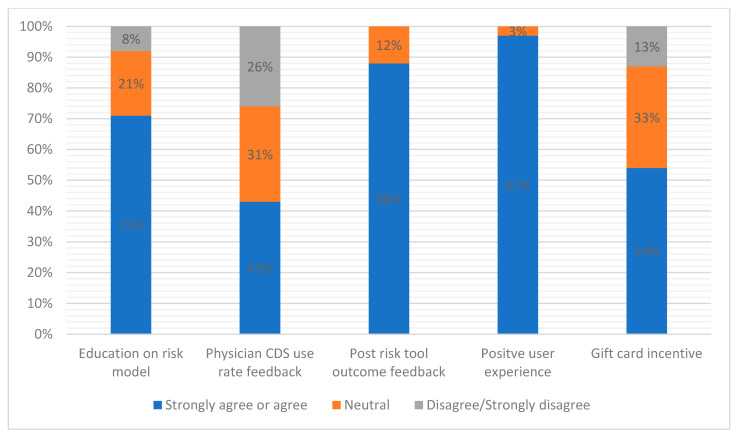
Survey data: ED providers’ impressions of key factors that would influence their likelihood of using an ML-based AHF risk model and paired CDS. Bars represent percentage of providers responding “agree/strongly agree”, “neutral”, or “disagree/strongly agree” to each potential motivator of use.

## Data Availability

Not applicable.

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
