# Peer review of "Barriers and Opportunities Regarding Implementation of a Machine Learning-Based Acute Heart Failure Risk Stratification Tool in the Emergency Department"

_diagnostics, 2022, doi:10.3390/diagnostics12102463_

Round 1

Reviewer 1 Report

This is an interesting manuscript discussing the barriers to the implantation of a machine-learning risk model and clinical decision support for ED patients with acute heart failure. I personally agree with this manuscript. The data are illustrated by figures and tables. The paper is well written.

Author Response

We appreciate this Reviewer's comments and support of the manuscript. 

Reviewer 2 Report

In the paper "Barriers and opportunities to implementation of a machine-learning based acute heart failure risk stratification tool in the emergency department" the authors conducted semi-structured interviews of eight front-line emergency department (ED) providers and surveyed 67 ED providers in order to understand barriers and opportunities to the successful implementation of a machine-learning based acute heart failure risk stratification tool. 

The research is well designed and conducted, the paper is well written and of interest to the reader

Author Response

We appreciate this Reviewer's comments and support.  

Reviewer 3 Report

In the abstract, the result of this work must be described briefly with data. The result of this work is not clear.

In the Introduction part, the strong points of this proposed method should be further stated and the organization of this whole paper is supposed to be provided at the end.

The problem definition of this work is not clear. Also, no related works.

Overall, the authors have made a good attempt. However, the authors' proposed method does not adequately describe their data. The results are not supported by any theoretical/mathematical reasons. Readers will fail to understand the scientific contribution of this research. The authors should justify the effectiveness of the proposed technique theoretically.

The results of this research are not clear in Conclusions. Furthermore, the benefits of the proposed method are not supported by theory. So, I fail to understand the scientific contribution of this research.

Author Response

Reviewer 3: In the abstract, the result of this work must be described briefly with data. The result of this work is not clear.

Author’s response: We appreciate the reviewer’s comment and will try to state the results more clearly in the Abstract.   Specifically, we plan to state:

“From surveys and interviews, providers communicated that decision support must be streamlined into workflows with minimal disruptions. When asked about what areas were most important for tailored clinical decision support, 98% and 79% of surveyed providers wanted personalized assistance with discharge planning and ED disposition decision-making, respectively.  Receiving feedback on patient outcomes after risk tool use was seen an opportunity to increase acceptance, and few providers (<10%) had significant hesitations with using a ML-based tool after education on its use. In conclusion, engagement with key front-line users on optimal design of the algorithm and decision support may contribute to broader uptake, acceptance, and adoption of recommendations with clinical decisions.”

In the Introduction part, the strong points of this proposed method should be further stated and the organization of this whole paper is supposed to be provided at the end.

Author’s response: Thank you for this comment.  In the last paragraph of the introduction section, we state: “The goal of this study was to explore factors influencing the integration and adoption of a ML-based risk prediction tool and paired CDS for ED patients with AHF into clinical workflows.”  We plan to add the follow this sentence with a sentence that provides a framework for the study design/methods of the manuscript:

“To achieve this goal, we conducted semi-structured interviews and surveys with front line ED-physicians and used a mixed-methods analysis approach to better understand barriers and opportunities to optimal implementation of the tool and paired CDS.”      

The problem definition of this work is not clear. Also, no related works.

Author’s response: We describe the challenge this study is trying to overcome in the last two paragraphs of the introduction  and cite relevant related works:

“Implementation of a new risk tool and clinical decision support (CDS) into a busy ED can be challenging [16–18]. EHR tools for patients with AHF have been successfully deployed, with limited impacts  observed when decision support was not added to prognostic scores [19,20].  Prior studies highlight an underappreciation of the steps needed to translate model predictions into real-time use and improved health [21,22]. Successful implementation requires careful consideration of various factors, including workflow adaptability [16], technical infrastructure [17,23], and clinician acceptance [16,17,24–26]. Some studies suggest lack of trust or familiarity with ML models may make their implementation more challenging [24,27].

While several studies describe the importance of establishing a framework for risk tool implementation [21–23,25,28,29], to our knowledge, none have focused on ED physician reported feedback to guide the planned implementation process of a ML-based tool paired with CDS in the ED setting… The goal of this study was to explore factors influencing the integration and adoption of a ML-based risk prediction tool and paired CDS for ED patients with AHF into clinical workflows. 

Overall, the authors have made a good attempt. However, the authors' proposed method does not adequately describe their data. The results are not supported by any theoretical/mathematical reasons. Readers will fail to understand the scientific contribution of this research. The authors should justify the effectiveness of the proposed technique theoretically.

Author’s response: We describe the methods and theoretical framework for our study in the first lines of the Methods section:

“Using a mixed-methods approach and conceptual model based on the Consolidated Framework for Implementation Research[21], we conducted semi-structured, in-depth video interviews with 8 ED providers and surveyed 103 ED providers.”

The results of this research are not clear in Conclusions. Furthermore, the benefits of the proposed method are not supported by theory. So, I fail to understand the scientific contribution of this research.

Author’s response:  We apologize that the importance of this study was not clear.  In the conclusion, we state the main results and implications of these findings:

“Our study findings from 8 interviewed and 67 surveyed ED physicians identify several important opportunities and barriers to implementation and real-time use of a ML-based risk model and CDS for ED patients with AHF.  Providers noted that personalized decision-support and use of risk-based pathways could help standardize hospital admission decisions and discharge planning. They highlighted the need for CDS to be integrated into workflows to minimize disruptions. Provider education prior to CDS implementation and regular feedback on patient outcomes after were seen as opportunities to increase acceptance and adoption.  Lessons learned from this study will be used to optimize development and implementation of AHF CDS in our health system and can be applied to other health systems considering integrating ML-based models into their EHR.”   

Reviewer 4 Report

I read the paper. It is well written, but I think that the methodology should be improved. Following some comments:

- It is difficult to interpret the results because they are qualitative presented. I suggest including scores to objectively quantified the answers. 

- I suggest applying statistical analysis in order to understand if the sample size is enough. 

- I suggest evaluating also the anamnestic data (age, experience in the field, etc.) of the clinicians. It can influence their opinions about machine learning. 

Author Response

I read the paper. It is well written, but I think that the methodology should be improved. Following some comments:

- It is difficult to interpret the results because they are qualitative presented. I suggest including scores to objectively quantified the answers. 

Author’s response:  As this is a mixed methods study with data from interviews (qualitative) and surveys (quantitative), we tried to present our findings in both qualitative (key quotations from interviews) and quantitative formats (bar graphs with percentages from survey data). 

- I suggest applying statistical analysis in order to understand if the sample size is enough. I suggest evaluating also the anamnestic data (age, experience in the field, etc.) of the clinicians. It can influence their opinions about machine learning

Author’s response: Thank you for this comment.  We sent the interview to all emergency department providers and had a 65% response rate, with 67 physicians sending back the survey.  A response rate greater than 50% is considered more than adequate for most surveys.  We present descriptive statistics of physicians’ responses to questions related to barriers and opportunities to implementation of the risk tool.  We did not perform any tests to show the statistical significance of correlation between physician characteristics and survey responses because of the small sample size and because we would not expect statistical associations between respondent characteristics and qualitative answers. 

Round 2

Reviewer 3 Report

The authors have addressed the reviewer's concerns and the revised version of the manuscript appears to be good.

Reviewer 4 Report

The authors solved my issues